# Communication between Mast Cells and Group 2 Innate Lymphoid Cells in the Skin

**DOI:** 10.3390/cells13050462

**Published:** 2024-03-06

**Authors:** Yeganeh Mehrani, Solmaz Morovati, Tahmineh Tajik, Soroush Sarmadi, Ali Bitaraf, Zahra Sourani, Mohammad Shahverdi, Helia Javadi, Julia E. Kakish, Byram W. Bridle, Khalil Karimi

**Affiliations:** 1Department of Pathobiology, Ontario Veterinary College, University of Guelph, Guelph, ON N1G 2W1, Canada; ymehrani@uoguelph.ca (Y.M.); jkakish@uoguelph.ca (J.E.K.); 2Department of Clinical Sciences, School of Veterinary Medicine, Ferdowsi University of Mashhad, Mashhad 91779-48974, Iran; 3Division of Biotechnology, Department of Pathobiology, School of Veterinary Medicine, Shiraz University, Shiraz 71557-13876, Iran; s.morovati@shirazu.ac.ir; 4Department of Pathobiology, School of Veterinary Medicine, Ferdowsi University of Mashhad, Mashhad 91779-48974, Iran; ta.tajik@mail.um.ac.ir; 5Department of Microbiology and Immunology, Faculty of Veterinary Medicine, University of Tehran, Tehran 14199-63114, Iran; soroush.sarmadi@ut.ac.ir; 6Experimental Medicine Research Center, Tehran University of Medical Sciences, Tehran 14167-53955, Iran; 7Department of Pathology, School of Veterinary Medicine, Shahrekord University, Shahrekord 88186-34141, Iran; sourani@stu.sku.ac.ir (Z.S.); shahverdi.m@skums.ac.ir (M.S.); 8Clinical Biochemistry Research Center, School of Medicine, Shahrekord University of Medical Sciences, Shahrekord 88157-13471, Iran; 9Department of Medical Sciences, Schulich School of Medicine & Dentistry, University of Western Ontario, London, ON N6A 3K7, Canada; hjavadi@uwo.ca

**Keywords:** MCs, ILC2s, skin, disease

## Abstract

The skin is a dynamic organ with a complex immune network critical for maintaining balance and defending against various pathogens. Different types of cells in the skin, such as mast cells (MCs) and group 2 innate lymphoid cells (ILC2s), contribute to immune regulation and play essential roles in the early immune response to various triggers, including allergens. It is beneficial to dissect cell-to-cell interactions in the skin to elucidate the mechanisms underlying skin immunity. The current manuscript concentrates explicitly on the communication pathways between MCs and ILC2s in the skin, highlighting their ability to regulate immune responses, inflammation, and tissue repair. Furthermore, it discusses how the interactions between MCs and ILC2s play a crucial role in various skin conditions, such as autoimmune diseases, dermatological disorders, and allergic reactions. Understanding the complex interactions between MCs and ILC2s in different skin conditions is crucial to developing targeted treatments for related disorders. The discovery of shared pathways could pave the way for novel therapeutic interventions to restore immunological balance in diseased skin tissues.

## 1. Skin Anatomy: Location of Mast Cells (MCs) and Group 2 Innate Lymphoid Cells (ILC2s)

As the body’s outermost organ, the skin acts as a mechanical and immunological barrier to protect against external threats through resident or infiltrating immune cells [1]. Mammalian skin has three layers: the epidermis, dermis, and hypodermis (subcutis), each with a unique anatomy and function [2]. The epidermis forms the outermost layer of the skin and serves as a protective barrier. It produces vitamin D and pigmentation [3,4]. Keratinocytes, which produce keratin and lipids, constitute 90-95% of the epidermis, while Langerhans cells, resident memory CD8^+^ T cells, melanocytes, and Merkel cells comprise the remaining 5% [5]. The middle layer of the skin, the dermis, lies beneath the epidermis and provides structural and nutritional support and contains various immune cells such as fibroblasts, MCs, macrophages, dendritic cells (DCs), CD4^+^ and CD8^+^ T cells, B cells, natural killer cells (NK cells), and ILCs [1,2,5]. Beneath the dermis lies the hypodermis [3], which contains many immune cells such as macrophages and T cells, and generates several mediators, including growth factors, adipokines, and cytokines [6].

MCs originate as progenitors (MCps) in the bone marrow and mature in peripheral tissues. Transcription factors regulate the process of maturation. Once committed to maturation, MCps express essential markers such as the c-kit receptor for SCF and the FcεRI receptor for IgE, similar to mature MCs [7]. Human MCs can be categorized into three types based on their expression of different proteins. MCTs express only tryptase, MCTCs contain tryptase, chymase, carboxypeptidase, and a cathepsin G-like proteinase, whereas MCCs express chymase only [8,9]. MCCs are predominantly found in the stomach and lungs, while MCTs are in the lungs and intestinal mucosa, and MCTCs are located in the subepithelial layers of various connective tissues, including the skin and gastrointestinal tract [10].

The ILC family comprises NK cells, ILC1, ILC2, and ILC3, and are predominantly tissue-resident lymphocytes found in barrier tissues such as the skin. They play a crucial role in immune responses against viruses, bacteria, parasites, and transformed cells [11,12,13].

MCs release IL-13, which directly stimulates the activity of innate lymphoid cell type 2 (ILC2s). These cytokines signal ILC2s about potential allergens or pathogens [14,15,16]. Similarly, ILC2s produce IL-5 and IL-13, which enhance local skin inflammation when activated by MC cytokines [17]. Additionally, both MCs and ILC2s play essential roles in regulating immune responses [17]. For example, MCs can promote or inhibit inflammation, affecting the behavior of nearby ILC2s and strengthening their responses. When working together, MCs and ILC2s can detect and respond to threats by regulating local immune responses.

## 2. Mechanisms of MC Activation in the Skin

MCTCs are the primary MC subtypes in the human skin [18] close to blood vessels and nerves. MCTCs can send signals alerting the immune system to the presence of harmful pathogens or other microenvironmental stimuli, thereby recruiting and activating innate and adaptive immune system members and commencing immunoregulatory processes [19]. In response to both IgE-dependent and IgE-independent stimuli, MCs release mediators [20,21]. These mediators include a variety of cytokines such as IL-1β, IL-4, IL-6, IL-8 (CXCL8), IL-10, MCP-1 (CCL2), and tumor necrosis factor (TNF), as well as vasoactive amines such as histamine and serotonin, growth factors like fibroblast growth factor 2 (FGF-2), platelet-derived growth factor (PDGF), transforming growth factor-β (TGF-β), vascular endothelial growth factor (VEGF), and proteases including chymase and tryptase. Additionally, MCs release proteoglycans like heparin and a range of lipid mediators, including platelet-activating factor (PAF), prostaglandin D2 (PGD2), and prostaglandin E2 (PGE2) [22,23]. Various inflammatory disorders in the skin involve significant MC activation, which will be briefly reviewed in the following sections.

### 2.1. IgE-Mediated Activation of MCs

MCs are characterized as a major player in allergic reactions and disease, which are mainly activated through an IgE-dependent mechanism that occurs through cross-linkage of the high-affinity IgE receptor (FcɛRI), expressed on the surfaces of MCs by allergens and pre-bound immunoglobulin (IgE) [24,25]. The activation of FcɛRI increases the intracellular calcium flux, intracellular granules, and the activation of transcription factors for eicosanoids, cytokines, and chemokines production [25]. After MCs degranulate, they release various mediators, including cytokines like IL-4. These cytokines play a crucial role in stimulating the production of IgE antibodies by B cells. The interaction between IgE and its receptor FcεRI on MCs increases FcεRI expression, creating a positive feedback loop. This feedback loop further highlights the complex interplay between MCs and B cells in the immune response [16]. Furthermore, Buttgereit et al. demonstrated that this method of MC activation seems to increase the mitochondrial oxygen consumption rate (OCR) and extracellular acidification. They also indicated that IgE-dependent MC activation enhances cytokine and growth factor secretion and degranulation [24].

In atopic dermatitis (AD), MCs were observed to be closely associated with endothelial cells, possibly leading to the stimulation of vascular proliferation. This change is believed to be mediated by MCs’ release of proangiogenic substances [26]. As previously mentioned, cutaneous MCs respond to the interaction between IgE and FcεRI by releasing soluble inflammatory mediators through degranulation. Additionally, skin MCs tend toward the preferential absorption of IgE from the bloodstream [27]. Histamine released following degranulation induces the prompt development of localized edema by vasodilation and the augmentation of vascular permeability [28]. Exposure to allergens can trigger widespread allergic reactions known as systemic anaphylaxis. This leads to MC activation and a significant release of various chemical mediators [29].

### 2.2. Non-IgE-Mediated Activation of MCs

Besides IgE receptors that trigger allergic reactions, MCs also express pattern recognition receptors like TLRs, Fc receptors, and complement receptors [19]. Moreover, their alarmin and purinergic receptors equip MCs to sense cell stress and tissue damage. MCs can also be activated or modulated by binding cytokines, growth factors, chemokines, and neuropeptides [16]. In this regard, the identification of Mas-related G protein-coupled receptor member X2 (MRGPRX2), or its mouse ortholog Mrgprb2, as a pseudo-allergic/neurogenic route of IgE-independent MC degranulation, in recent years has changed the former view in MC biology [30]. MRGPRX2 is reportedly activated by the plethora of endogenous and exogenous ligands, such as antimicrobial host defense peptides (cathelicidin and β-defensins), neuropeptides (such as substance P and somatostatin), eosinophil granule proteins, and many US Food and Drug Administration-approved peptidergic drugs known to trigger pseudo-allergic reactions in susceptible individuals [31,32]. Cutaneous MCs are the most abundant MC population in the body [16], express the highest levels of MRGPRX2, and respond most vigorously to its ligands [33]. The activation of MRGPRX2 triggers G protein-coupled signaling cascades. This induced phospholipase C signaling, calcium release, and degranulation in MCs [32]. In addition to degranulation, MRGPRX2 can elicit cytokine responses in skin MCs [34].

The activation of MCs via MRGPRX2 may contribute to neurogenic inflammation, pain, itching, and pruritic skin diseases, including chronic spontaneous urticaria (CSU) and atopic dermatitis [32]. After activation and degranulation, MCs release inflammatory, algogenic, and pruritogenic mediators, which bind to specific nociceptors on sensory nerve fibers and initiate reciprocal communication. Activated nerve fibers, in turn, release inflammatory and vasoactive neuropeptides such as SP, which are MRGPRX2 agonists, further promoting the vicious cycle toward disease exacerbation [35]. In this regard, increased numbers of MRGPRX2-positive cells have been reported in the skin of patients with CSU [36]. Moreover, recent studies have suggested pseudo-allergic drug reactions may occur via MRGPRX2. In this line, a study by Hao et al. showed that the application of imiquimod, an FDA-approved drug widely used in dermatology, induced dermatitis with inflammatory cell infiltration plus MC activation in the skin of wild-type mice but not in MRGPRB2−/− mice [37].

IgE-independent mechanisms of MC activation have been recently explained in physical urticaria. Boyden et al. found a specific mutation in the autoinhibitory subunit of mechanical-sensing ADGRE2 (adhesion G protein-coupled receptor E2) in patients with vibratory urticaria. The ADGRE2 receptor undergoes autocatalytic cleavage, and its extracellular subunit binds a transmembrane subunit. The variant probably destabilizes an autoinhibitory subunit interaction sensitizing MCs to IgE-independent vibration-induced degranulation [38].

For a rapid response to infection or cellular damage, MCs express pattern recognition receptors (PRRs) that recognize pathogen-associated molecular patterns (PAMPs) in bacterial, viral, fungal, or parasitic components and damage-associated molecular patterns (DAMPs) in host-derived molecules. Five families of PRRs include TLRs, C-type lectin-like receptors (CLRs), retinoic acid-inducible gene I (RIG-I)-like receptors (RLRs), nucleotide-binding oligomerization domain (NOD)-like receptors (NLRs) [39].

MCs recognize bacteria through TLR signaling [40]. For instance, Lei et al. reported that MCs were activated in an *S. aureus*-infected skin abscess murine model, and MC-derived tryptase contributed to the inflammation [41] However, the exact mechanism of MC activation was not investigated in that study.

CLRs are a group of PRRs with vital importance in antifungal defense. Dectin-1 is the best-known receptor in the whole family [39]. The binding of the CLR dectin-1 to the fungi Malassezia sporodialis component curdlan leads to MC degranulation and the release of leukotriene C4, IL-6, and CCL2 [42]. It is believed that Malassezia sporodialis plays a role in the development of atopic dermatitis [43].

MCs sense cell stress and keratinocyte (KC) damage via concomitant IL-33/ST2, ATP/P2X7, and thymic stromal lymphopoietin (TSLP)/TSLP receptor signaling. In itching skin diseases, where itching and scratching promote KC activation, skin infections, and inflammation, activated KCs release substances such as TSLP and IL-33, activating MCs through their specific receptors. TSLP can also promote MRGPRX2 signaling [16,32,44]. In contrast with degranulation-competent receptors like FcɛRI and MRGPRX2 that induce a calcium signal, ST2 and TSLPR are incapable of inducing a Ca2^+^ pulse and, consequently, do not lead to degranulation, even though they can influence the degranulation response induced by other receptors [45].

## 3. Activation Mechanisms of ILC2s in the Skin

Each subtype of ILCs demonstrates unique functional properties and generates distinctive effector molecules when encountering pathogens [46]. ILC2s are significant in immunosurveillance, immune regulation, and skin wound healing.

They can also interact with MCs and suppress IgE-dependent cytokine release from MCs by producing IL-13 [47]. The activation of ILC2s occurs by epithelial cytokines such as IL-33, IL-25, and thymic stromal lymphopoietin (TSLP). These activations also occur by T lymphocyte-derived cytokines, including IL-2, IL-4, IL-7, and IL-9, and lipid mediators, PGD2, and cysteinyl leukotrienes [48]. The main cytokines produced by ILC2s are IL-5, IL-9, and IL-13, but they can also produce IL-6, IL-10, granulocyte–macrophage colony-stimulating factor (GM-CSF), epidermal growth factor-like molecule amphiregulin, and small amounts of IL-4. Cytokines play crucial functions in the defense against helminth infections and in regulating allergic reactions [48,49,50]. ILC2s are considered to be effective cells in responding to parasites and allergens and initiating type 2 immunity, which is mediated by type 2 cytokines to recruit and activate eosinophils to eliminate the invading pathogens [51] and provide protection against helminths and toxins and contribute to tissue repair [50]. ILC2s seem to provoke and develop an appropriate Th2 response directly by the expression of major histocompatibility complex class II (MHC-II), and they can also indirectly promote Th2 differentiation [52]. In addition to MHC-II, ILC2s can directly provoke CD4^+^ T cells by expressing OX40L, CD80, and CD86 [48]. ILC2s are critical in the immune response to irritants, including allergens. When activated, ILC2s generate significant amounts of IL-5 and IL-13, developing type 2 inflammation [53].

### Cytokine-Mediated Activation of ILC2s

Cytokines can activate ILC2s that various immune cells generate, including keratinocytes and DCs. For instance, IL-33 is a distinctive cytokine that serves a crucial function in damaged or infected tissues, and it binds to its receptor serum stimulation-2 (ST2) on ILC2s, leading to the activation and subsequent release of cytokines like IL5 and IL13 [54,55]. In addition, a recent study revealed that ILC2 activation by IL-25 is essential for IL-13 expression at sites of allergic skin inflammation [56].

## 4. The Multifaceted Roles of MCs and ILC2s in Skin Homeostasis and Immune Responses

### 4.1. Tissue Microenvironment

The local tissue microenvironment plays a crucial role in ILC activation. ILCs exhibit a remarkable degree of plasticity, enabling them to easily alter their phenotypic characteristics and functions in response to signals originating from their surroundings [57,58]. Various factors, such as oxygen levels, metabolic changes, or DAMPs in response to injury or infection, can trigger the activation of ILCs [59]. MCs are well-known for their ability to produce a wide range of effector chemicals, including histamine, which is a primary mediator stored in granules within MCs and is rapidly released upon stimulation [60]. MCs also generate and secrete pro-inflammatory cytokines such as TNF-α, VEGF, FGF 2, IL-1, IL-6, and IL-8 [61]. These chemicals serve as complex signaling agents, directing other immune cells to the precise location of infection or injury, thus initiating an inflammatory state and an appropriate immunological response [62,63]. In addition, MCs are responsible for producing chemotactic factors such as leukotrienes and PAF. These substances have the notable capacity to attract and stimulate other immune system cells [64]. Leukotrienes are important in allergic asthma since they contribute to bronchoconstriction, increased vascular permeability, and mucus secretion. In contrast, it has been observed that PAF plays a role in enhancing the recruitment of neutrophils and eosinophils to the site of inflammation [62].

MCs can also produce diverse proteases, including tryptase, chymase, carboxypeptidase A3, cathepsin G, granzyme B, matrix metalloproteinases, and renin. During degranulation, tryptase acts as an indicative marker for MC activation and can also directly cause damage to the invading pathogens. It also contributes to tissue remodeling in cases of inflammation by dermal fibroblast proliferation and the production of type I collagen [63,65,66]. Furthermore, MCs can generate lipid mediators, like prostaglandins and leukotrienes, through the activity of phospholipases. Prostaglandins play a role in regulating tone, platelet aggregation, and immune cell function. Leukotrienes are lipid mediators that participate in bronchoconstriction vasodilation, increased vascular permeability, and chemotaxis [67].

### 4.2. Physiological Functions in Normal and Healthy Situations

The ear skin of mice contains up to 10% MCs, the highest percentage of MCs in leukocyte populations [68]. Dermal MCs, with an estimated density of 7000 to 20,000 per cubic millimeter of skin, are commonly found near blood vessels, nerves, and lymphatics [69]. Over the past two decades, due to their widespread distribution in all tissues, evidence indicates that MCs are crucial for several physiological processes, such as skin barrier homeostasis, angiogenesis, and innate and adaptive immunity [63]. MCs use cytokines, chemokines, and growth factors to regulate skin homeostasis and barrier function by communicating with immune and non-immune cells nearby [70,71]. Fibroblasts, keratinocytes, and endothelial cells contribute to skin homeostasis through bidirectional interactions with MCs [16]. MCs stimulate fibroblast proliferation, while fibroblasts, in turn, control MC activation by releasing specific mediators to maintain skin barrier homeostasis [72]. MCs have both inhibitory and activating effects on keratinocytes by expressing different mediators. Bidirectionally, SCF production in keratinocytes triggers the differentiation of dermal MCs [73]. MCs enhance angiogenesis by impacting blood and lymphatic endothelial cells via several angiogenesis-related factors such as VEGF and histamine [70]. Activation of TLR2 and TLR4 by bacteria prompts MCs to release cytokines, which recruit neutrophils and NK cells to eliminate bacteria [74,75]. Cathelicidins, defensins, and psidins are antimicrobial agents effective against various microorganisms, including Gram-positive and Gram-negative bacteria, viruses, and fungi. These agents are naturally produced by MCs [76,77]. Additionally, interactions between MCs and DCs or T cells can influence the functions of these immune cells, thereby promoting adaptive immune responses [78,79].

Parallel to MCs, various subsets of ILCs have been identified in healthy skin. However, their precise physiological function is yet to be fully understood. ILCs are crucial in maintaining the skin barrier by producing cytokines and growth factors that affect epithelial cells [11]. Among these, ILC2s are the most abundant in the skin, and they produce amphiregulin, an epidermal growth factor, to support the repair of epithelial tissue [80]. ILC2s are located alongside skin-resident MCs in healthy skin to support their pro-inflammatory function. In homeostasis conditions, ILC2s produce IL-13, suppressing MCs’ function [81]. ILCs also help regulate T cells in the skin, essential for maintaining skin homeostasis. Interestingly, studies have shown that a lack of ILC2s in the skin leads to a decrease in the presence of regulatory T cells [82].

ILC2s obtained from peripheral blood and have skin-homing markers like CLA and CCR10 display an increase in skin-resident ILC2 levels ranging from 0.9% to 84%. This indicates that ILC2s, like other organs, originate from bone marrow, travel through the blood, and migrate to the skin [83].

## 5. Cross-Talk between Skin MCs and ILC2s

MCs and ILC2s have been detected in both human and mouse skin [72]. When activated via specific receptors, these two cell types release IL-4, IL-5, IL-13, and amphiregulin, showing similar biological activity. The simultaneous activation of MCs and ILC2s can potentially enhance allergic inflammation [72]. However, as ILC2s are located near MCs, they might receive MC-derived activation signals from them and vice versa.

ILC2-expressed IL-13 can revoke IgE-dependent cytokine release by MCs and act as a negative regulator for IgE-mediated responses [73]. IL-33-driven upregulation of IL-13 in MCs downregulates the IL-12 production by skin DCs, which restrains their ability to polarize naive CD4^+^ T cells into Th1 cells [74]. Moreover, in keratinocytes, MC-derived IL-4 and IL-13 decrease epithelial cadherin (E-cadherin) expression [75]. E-cadherin ligation negatively regulates ILC2s’ activity since it inhibits IL-5 and IL-13 production from human ILC2 via KLRG1 [76]. Taken together, ILC2-expressed IL-13 inhibits MC activation and alleviates the inflammatory response. In contrast, MC-released IL-13 stimulates ILC2s and promotes Th2 responses in AD.

In contrast with IL-13, ILC2s-released IL-5 may have an indirect stimulatory effect on MCs since it promotes the recruitment of eosinophils [73]. Eosinophil-released major basic protein (MBP) and eosinophil peroxidase (EPO) can cooperate with Mas-related G protein-coupled receptor member X2 (MRGPRX2) on MCs and activate them in an IgE-independent manner. In contrast, Kim et al. recently found that the MC production of IL-5 suppresses the activation of IL-13-producing ILC2s via maintaining the population of IL-10-expressing regulatory B cells [77].

IL-33-stimulated ILC2s also release IL-9, which, along with Th9-expressed IL-9, activates MCs to release IL-2. It has been shown that the activation of ILC2s is closely related to the presence of IL-2, and even human ILC2s can be activated only in the presence of IL-2. Surprisingly, IL-2 promotes ILC2s to self-release IL-9, critical in ILC2s’ survival and activation. In parallel with this positive loop, activated ILC2s enhance the activation of Th9 cells and lead to the further release of IL-9 [78,79] (Figure 1). However, MC-derived IL-2 was found to limit ILC2 activation via Treg stimulation and the release of IL-10 in mouse lung studies. Further investigation is required to determine the link between this discovery and skin inflammation [80].

Leyva-Castillo et al. found that ILC2 activation by keratinocyte-derived IL-25 also promotes the release of IL-13 at sites of allergic skin inflammation [56].

ILCs can differentiate from one subset into another due to their high plasticity potential, leading to changes in ILC composition in tissues. For example, Bal et al. showed that ILC2s could molecularly switch into IFN-γ-producing ILC1s under the influence of IL-1β and IL-12 in respiratory-related disorders, which could be reversed by IL-4 [81]. The study revealed the critical roles of IL-12 and IL-4 in determining the ratio of ILC2s to ILC1s in inflamed tissues. For example, in leishmaniasis, MC-deficient mice experience severe disease that contributes to the decreased infiltration of DCs, leading to a deficit in IL-12 and an impaired Th1 response [82]. In another study, localized cutaneous leishmaniasis was correlated with a higher proportion of ILC1s, while the disseminated form was accompanied by the majority of ILC2s [83]. It could be suggested that MCs may have a prominent role in determining the composition of ILCs in this disease, which requires more investigation.

In conclusion, the complex cross-talk between MCs and ILC2s in the skin reveals their multifaceted roles in modulating immune responses. Understanding the dynamic interaction between these cell types is crucial for developing targeted treatments for allergic skin disorders and related conditions.

### 5.1. Mastocytosis

Systemic mastocytosis is a hematological disorder characterized by the abnormal accumulation of MCs resulting from mutations in the KIT receptor. There is evidence regarding MC–ILC2s crosstalk, which can contribute to cutaneous local inflammation. A recent study compared patients diagnosed with systemic mastocytosis with healthy individuals [84]. According to the study, regardless of their serum tryptase levels, patients with skin lesions exhibited higher amounts of ILC2s in their peripheral blood. This was observed while the frequency of KIT^+^ILC2 precursors was comparable among all groups. The authors proposed that D816V^+^ MCs in the skin discharge some inflammatory mediators, including IL-1β, TGFβ, and IL-33-activating proteases, which activate local ILC2s and foster an environment that attracts circulating ILC2s.

The stimulated ILC2s may contribute to skin issues by synthesizing inflammatory substances, such as IL-9, and by enhancing the secretion of mediators by MCs. Additionally, other studies have demonstrated that prostaglandin D2 derived from MCs can promote cytokine secretion and the trafficking of ILC2s through the chemoattractant receptor-homologous molecule expressed on TH2 cells (CRTH2) receptor, which, then, may dampen MC activation [85]. Notably, CCR10 is expressed by KIT^+^ ILC2s, suggesting that this subgroup of ILC2s can migrate toward the skin. Moreover, the higher expression of soluble SCF in the affected skin of patients with mastocytosis indicates a potential interaction between SCF and KIT^+^ ILC2s [86] (Figure 2).

### 5.2. Atopic Dermatitis

ILC2s are more abundant in AD injuries and express ST2, IL-17RB, and TSLP-R more strongly, suggesting that they have an active phenotype. IL-4, IL-5, and IL-13 expression in these cells is subsequently upregulated in response to PGD2 discharged by MCs and other cells, as well as IL-33, IL-25, and TSLP released by keratinocytes. IL-33, released from damaged KCs, can act on MCs and ILC2s to enhance their activity and production of type 2 cytokines. MCs can also release IL-33 following IgE crosslinking and IL-33 binding to its receptor (ST2) in a feed-forward manner. In ILC2s, IL-33, in combination with ATP, is released from the damaged KCs and acts via the P2X7 receptor, triggering the production of IL-13. MCs also use the concomitant IL-33/ST2 and ATP/P2X7 signaling pathways to sense cell stress and tissue damage [16,87].

Also of interest is that while ILC2 numbers are increased in the lesions of AD patients, the quantity of these cells in circulation is similar in healthy individuals and those with atopic conditions [76]. However, the elevation in the number of ILC2s in the skin over 26 h cannot be solely attributed to local proliferation, which suggests that ILC2s can be recruited from the peripheral blood in response to allergen stimulation.

The proximity between ILC2 cells and MCs in the skin suggests that a mutual influence exists between these two cellular populations. Dermal ILC2s (dILC2s) exhibit a crucial role in the development of cutaneous inflammation such as AD [88]. dILC2 cells demonstrate a distinctive pattern of migration, characterized by intermittent bursts of movement accompanied by prolonged pauses. These pauses in migratory activity appear to result from interactions with MCs residing in the skin. Using intravital imaging, Roediger et al. revealed that treatment of mice lacking the recombination-activating gene 1 (Rag1−/−) with IL-2 as a T-cell-dependent regulator resulted in an expansion of stimulated dILC2 cells capable of releasing IL-5. This process finally led to the development of spontaneous dermatitis. In contrast, the authors observed that these cells exhibited an inhibitory interplay with MCs through IL-13. Moreover, they found that while the dILC2s could not generate IL-4 in normal physiological states, these cells gained the capability to produce IL-4 when exposed to TSLP-induced stimulation [88].

Based on their phenotype and activity, dILC2s manifest two distinct functional states: steady-state (naive) and stimulated dILC2s. Steady-state ILC2s exert an immuno-regulatory function by generating IL-13, modulating the activity of skin-resident MCs. IL-13 inhibits MCs’ IgE-dependent production of pro-inflammatory cytokines in a dose-dependent manner. The viability of this subset relies on IL-7. Meanwhile, stimulated dILC2s adopt a pro-inflammatory state and facilitate the infiltration of eosinophils and the activation of MCs, identified by a heightened secretion of IL-5 [89]. Indeed, the granules found in MCs, containing chymase and tryptase, play a role in facilitating the maturation of IL-33, leading to the activation and expansion of ILC2s [89,90] (Figure 3). These mechanisms potentially sustain inflammation within the skin.

### 5.3. Systemic Sclerosis

Signals from IL-4, TGF-β, and TSLP influence the polarization of Th9 cells [91]. Additionally, OX40 ligation, when combined with TGF-β and IL-4, can convert CD4^+^ conventional T cells to Th9 cells [92]. Guggino et al. investigated the involvement of Th9 in the development of systemic sclerosis (SSc) [93], demonstrating the upregulation of the IL-9/IL-9R axis components, including IL-9, IL-9R, IL-4, TSLP, and TGF-β, in the inflamed skin of SSc individuals. Th9 cells were recognized as the main source of IL-9 in the skin. The number of Th9 cells showed a positive association with the modified Rodnan skin score, indicating their potential involvement in skin manifestations of SSc. In addition, IL-9 receptor expression was increased in recruited MCs, neutrophils, and mononuclear cells. Over-expression of IL-9 was surprisingly related to the expansion of ILC2s and MCs in SSc patients [93] (Figure 4). In more detail, MCs isolated from the peripheral blood showed significant expansion upon exposure to IL-9 and increased production of IL-2. This, in turn, induced the activation and proliferation of ILC2s. These findings suggest that targeting the pathway involving MCs, ILC2s, and IL-9 may be a potential therapeutic intervention for SSc patients.

### 5.4. Chronic Spontaneous Urticaria

Chronic spontaneous urticaria (CSU) is characterized by unexpected and impulsive wheal and angioedema development [94]. Several significant mechanisms of MC degranulation have been identified, particularly in CSU. Notably, the activation of Mas-related G protein-coupled receptor X2 (MRGPRX2) is observed in the skin MCs of patients suffering from CSU. Other pathways contributing to MC activation in CSU involve IL-33, ST2, and thymic stromal lymphopoietin (TSLP) [95]. It has been demonstrated that the amounts of IL-25, IL-33, and TSLP are elevated in the affected skin of patients with CSU, indicating the potential involvement of ILC2s in the pathological processes of CSU [96].

Studies suggest that VEGF plays a crucial role in allergic diseases, including CSU, and it is considered a potential blood biomarker for detecting CSU. MCs have been identified as one of the primary sources of VEGF. In vitro research reveals that the serum of CSU patients activates MCs, resulting in the release of VEGF through the PI3K/Akt/p38 MAPK/HIF-1α axis. Notably, vitamin D has been found to prevent the activation of this axis, leading to reduced VEGF synthesis in response to stimulation. This unveils a potential mechanism by which vitamin D therapy could benefit CSU patients [97]. It is also interesting to note that a significant proportion of patients with CSU have been found to exhibit vitamin D deficiency [98]. Vitamin D can inhibit the activity of ILC2s [98]. Hence, vitamin D insufficiency may play a significant role in the pathogenesis of CSU. The significant role of ILC2s in promoting allergic responses makes them a potentially intriguing target for future therapeutic interventions in CSU (Figure 5) [99].

In conclusion, ILC2s interact with MCs to contribute to the pathophysiology that develops in CSU. Strategies to reestablish immunological balance in the skin can be developed by looking at the changes in ILCs and MC-derived cytokines in the afflicted skin tissues during CSU.

## 6. Conclusions

The interaction between MCs and ILC2s in the skin is a complex interplay with positive and negative effects on immune responses. MCs are essential components of the immune system that are responsible for allergic reactions. Meanwhile, ILC2s significantly drive type 2 innate immune responses and are associated with inflammatory skin diseases and autoimmune-related conditions [99]. Skin disorders involve a complex interaction between MCs and ILCs through cytokines and receptors. This interaction significantly affects the development of skin diseases. For instance, in mastocytosis, MCs activate ILC2s, which leads to skin inflammation. A potential pathway involving MC–ILC2–Th9 in systemic sclerosis could be targeted for therapy. Atopic dermatitis shows higher activity of ILC2s, which are influenced by MC signals. Chronic spontaneous urticaria is characterized by increased levels of IL-25, IL-33, and TSLP, indicating the possible involvement of ILC2s. These interactions significantly contribute to disease development and offer promising avenues for future research and therapeutic interventions in skin conditions. To better understand skin immunity and develop targeted treatments, further research is necessary to determine the exact mechanisms governing these interactions.

## Figures and Tables

**Figure 1 cells-13-00462-f001:**
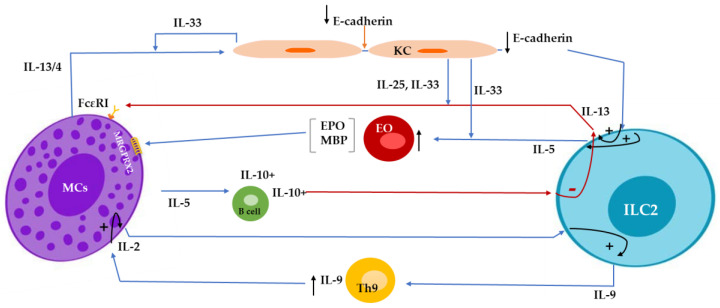
Mast cell (MC)-derived IL-4/IL-13 decreases E-cadherin expression in keratinocytes and subsequently attenuates E-cadherin-mediated suppression of IL-5 and IL-13 production from innate lymphoid cell type 2 (ILC2s). IL-33- and IL-25-stimulated ILC2s release IL13, a negative regulator for IgE-mediated responses in MCs. ILC2s-released IL-5 promotes the recruitment of eosinophils, which produce major basic protein (MBP) and eosinophil peroxidase (EPO) and, subsequently, stimulate MRGPRX2 (Mas-related G protein-coupled receptor member X2) on MCs. MC-released IL-5, maintaining the population of IL-10-expressing regulatory B cells, suppresses the activation of IL-13-producing ILC2s. ILC2s release IL-9, enhance the activation of Th9 cells, and, along with Th9-expressed IL-9, activate MCs to release IL-2, promoting ILC2 to produce more IL-9, resulting in a positive loop.

**Figure 2 cells-13-00462-f002:**
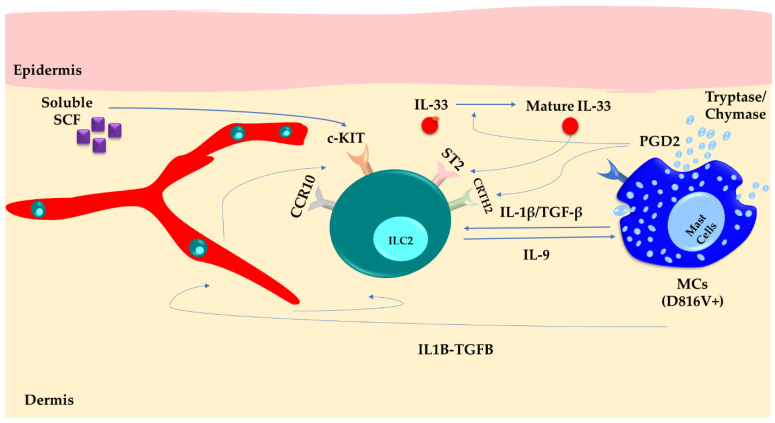
Skin mast cells (MCs) carrying the D816V+ mutation produce inflammatory mediators like IL-1β, TGF-β, PGD2, chymase, and tryptase proteases. This constant secretion activates ILC2s within the skin, creating a favorable environment that attracts circulating ILC2s through the CCR10 pathway. As a result, the persistent activation of ILC2s leads to the production of IL-9. This not only amplifies MC activity and skin symptoms but also facilitates the trafficking of ILC2s into the bloodstream.

**Figure 3 cells-13-00462-f003:**
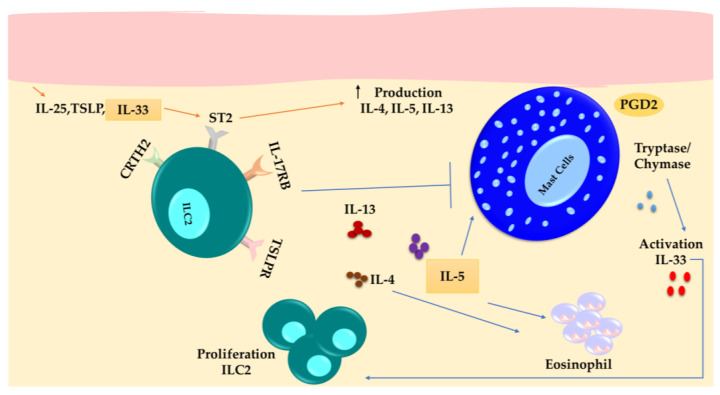
ILC2s are more abundant in the lesions of individuals with atopic dermatitis. ILC2s exhibit increased expression of ST2, IL-17RB, and TSLP-R. They produce IL-13, IL-5, and IL-4 in response to signals from keratinocytes such as IL-33, IL-25, and TSLP, as well as PGD2 expressed by MCs. IL-5 activates eosinophils and MCs, while IL-13 suppresses MCs.

**Figure 4 cells-13-00462-f004:**
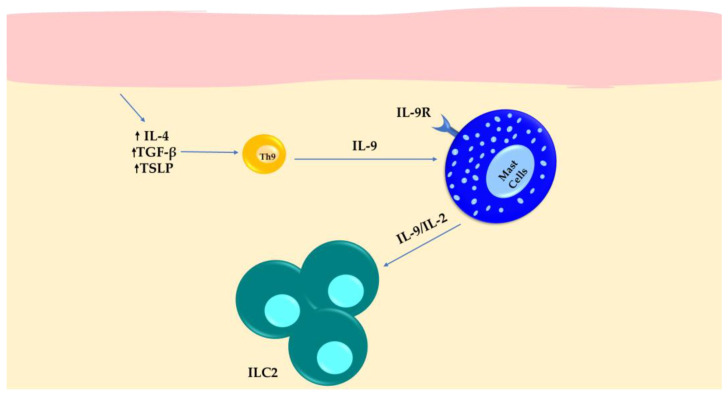
The expression of IL-9 in Th9 cells and IL-9 receptor in MCs results in the expansion of ILC2s in SSc patients with inflamed skin.

**Figure 5 cells-13-00462-f005:**
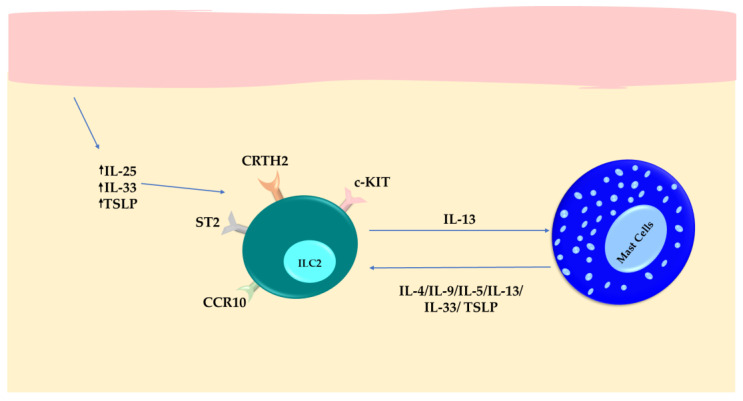
Inflammatory responses in patients with chronic spontaneous urticaria (CSU). Upregulation of IL-25, IL-33, and TSLP in keratinocytes contributes to the expansion of local ILC2s. The activated ILC2s then increase the expression of IL-13, which has a role in regulating MCs. In turn, MCs prolong the longevity of ILC2s and help to maintain cytokine secretion by their inflammatory mediators.

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
