# Peer review of "Communication between Mast Cells and Group 2 Innate Lymphoid Cells in the Skin"

_cells, 2024, doi:10.3390/cells13050462_

Round 1

Reviewer 1 Report

Comments and Suggestions for Authors

l60 : not clear, precise leave the BM as Mast cell-committed progenitors (MCPs) (ref) and perhaps mention that they can have an embryologic origin (Dahlin work) and https://www.cell.com/immunity/pdfExtended/S1074-7613(18)30438-2

l62 precise human mast cell

l72 IL-25 is not a typical MC cytokine even if published in mouse in 2003

l85 unclear sentence “MCs respond to various stimuli, releasing biologically active IgE-dependent 85 and independent mediators (20, 21)”

l87 il-1b, IL-4 and not IL4 ect, TNF and not TNFalpha (renamed)

l92 in fact MCTC do not produce cysteinyl leucotrienes (MCT yes)

l103 it is not degranulation that activates B cells and… it is IL-4

l116-117 not clear

l120 “In addition to expressing the IgE receptors that cause allergies” it is not the receptor that cause allergies, needs rephrasing

l120-137 lack of  references

l256-276 these paragraphs could appear earlier or combined with previous paragraphs because some points are already mentioned it could be interesting to mention the proportion of ILC2 and MCs in the skin (ex % total leukocytes)

l413 “The polarization of Th9 cells is influenced by signals received by IL-4, (TGF)-β, and 413 TSLP (97).” Unclear, add OX40L role in this polarization

some typos ex naïve

Comments on the Quality of English Language

some sentences need rephrasing

generally correct

Reviewer 2 Report

Comments and Suggestions for Authors

This review article summarizes recent papers regarding mast cells and ILC2 in the skin and their mutual interaction. The information included in this manuscript is reliable and will be useful for readers of this journal. I have some minor comments on the typographical errors and inconsistent writings.

I have both major and minor comments.

Major comment.

Lines 299 to 304.

Activated MCs release IL-2, which in turn activates ILC2. However, the mechanisms may be more complicated; MC-derived IL-2 can also limit ILC2 activation indirectly, via Treg stimulation and release of IL-10 in mouse lung study (Morita H, et al. Immunity 43:175,2015). Although we do not know whether this finding can be applied to skin inflammation, it would be better to mention this.

Lines 81, 98, 120, 183, 205, 242, 256, 265, 270 and others.

Indentation of sentences needs to be uniform.

Lines 87, 413, 416 and Figure 2

The names of cytokines need to be checked and corrected when necessary. For example, ILb, IL4, IL6, IL8, IL10 (line 87) should be changed to IL-1b, IL-4, IL-6, IL-8, IL-10. (TGF) (lines 413 and 416), and IL-1B-TGFB should be checked by the authors.

Figure 1.

The characters and arrows are small and thin. They should be emphasized.

Comments on the Quality of English Language

None.
